# Towards explainable interaction prediction: Embedding biological hierarchies into hyperbolic interaction space

**Domonkos Pogány**[ORCID]*, **Péter Antal**

Department of Measurement and Information Systems, Budapest University of Technology and Economics, Budapest, Hungary

* pogany@mit.bme.hu

## Abstract

Given the prolonged timelines and high costs associated with traditional approaches, accelerating drug development is crucial. Computational methods, particularly drug-target interaction prediction, have emerged as efficient tools, yet the explainability of machine learning models remains a challenge. Our work aims to provide more interpretable interaction prediction models using similarity-based prediction in a latent space aligned to biological hierarchies. We investigated integrating drug and protein hierarchies into a joint-embedding drug-target latent space via embedding regularization by conducting a comparative analysis between models employing traditional flat Euclidean vector spaces and those utilizing hyperbolic embeddings. Besides, we provided a latent space analysis as an example to show how we can gain visual insights into the trained model with the help of dimensionality reduction. Our results demonstrate that hierarchy regularization improves interpretability without compromising predictive performance. Furthermore, integrating hyperbolic embeddings, coupled with regularization, enhances the quality of the embedded hierarchy trees. Our approach enables a more informed and insightful application of interaction prediction models in drug discovery by constructing an interpretable hyperbolic latent space, simultaneously incorporating drug and target hierarchies and pairing them with available interaction information. Moreover, compatible with pairwise methods, the approach allows for additional transparency through existing explainable AI solutions.

## Introduction

Modern drug development, despite its critical role, is characterized by lengthy timelines and high costs, with bringing a new drug to market often requiring 10–15 years and 1.5–2.0 billion USD [1]. A promising strategy to accelerate drug development is *drug repositioning*, involving the use of existing drugs for new therapeutic applications. Computational methods, especially those leveraging *drug-target interaction* (DTI) prediction, have been widely used due to advancements in machine learning and the availability of vast databases. These approaches offer efficiency and accuracy, reducing the time and cost of experimentally measuring the

**Data Availability Statement:** All utilized datasets have been previously published and are freely available. The two interaction datasets, KIBA and NURA are available free of charge via the following Zenodo open repositories: https://zenodo.org/

records/5105698 (Making sense of large-scale kinase inhibitor bioactivity data sets: a comparative and integrative analysis; DOI: 10.5281/zenodo. 5105698) and https://zenodo.org/records/3999420 (NUclear Receptor Activity (NURA) dataset; DOI: 10.5281/zenodo.3999420). The used extra hierarchical information is available through the corresponding papers: NR (DOI: 10.1093/bib/ bbac351), Kinase (DOI: 10.1126/science. 1075762), and ATC (DOI: 10.1021/acs.jcim. 0c00681).

**Funding:** The project supported by the Doctoral Excellence Fellowship Programme (DCEP) is funded by the National Research Development and Innovation Fund of the Ministry of Culture and Innovation and the Budapest University of Technology and Economics, under a grant agreement with the National Research, Development and Innovation Office (2020-2.1.1-ED-2023-00239) (PD). This research was also funded by the J. Heim Student Scholarship (PD), the OTKA-K139330, the European Union (EU) Joint Program on Neurodegenerative Disease (JPND) Grant: (SOLID JPND2021-650-233), the National Research, Development, and Innovation Fund of Hungary under Grant TKP2021-EGA-02, the European Union project RRF-2.3.1-21-2022-00004 within the framework of the Artificial Intelligence National Laboratory. The funders had no role in study design, data collection and analysis, decision to publish, or preparation of the manuscript.

**Competing interests:** The authors have declared that no competing interests exist.

interactions [2, 3]. Nevertheless, besides predictive performance, the interpretability of DTI models is another important aspect that needs to be addressed.

Another prominent area in machine learning research is the exploration of hyperbolic embeddings, recognizing their potential in handling datasets with intrinsic hierarchies and non-linear structures more effectively than traditional Euclidean representations. Incorporating hyperbolic geometry into machine learning models allows hierarchical data representation with less distortion [4]. Recently, hyperbolic methods have gained more interest in DTI prediction and repositioning due to their ability to capture underlying biological structures and achieve enhanced predictive performance with lower dimensions [5–7]. While some studies explore embedding drug hierarchies in hyperbolic space [8], to the best of our knowledge, as for now, no known solutions are integrating both drug and target hierarchies within a DTI model.

Our work aims to incorporate information from drug and target hierarchies into DTI prediction models to enhance explainability by constructing a meaningful latent space. Leveraging distance-based DTI predictors for their inherent interpretability, our approach regularizes the *joint-embedding* drug-target latent space, making it simultaneously embed drug and target hierarchies while pairing them based on the available interaction information. We explore the effect of the hierarchy-based embedding regularization on both Euclidean and non-Euclidean manifolds. Additionally, we conduct a latent space analysis to gain a deeper understanding of the resulting drug-target model space, providing insights into the organization and interpretability of hierarchical information.

In summary, our contributions include introducing a method to embed drug and target hierarchies into a shared DTI representation space, evaluating the impact of prior hierarchy regularization on DTI prediction models, and offering a comprehensive latent space analysis as an illustrative example to interpret the resulting model. As a result, our joint-embedding hierarchy regularization leads to a more interpretable and meaningful latent space with only a slight performance cost, facilitating a more informed and insightful application of DTI prediction models in drug discovery. An overview of the proposed method is provided in Fig 1.

The paper proceeds as follows: Section Preliminaries provides background information, Section Methods outlines our datasets and methods, Section Results presents the outcomes of our comparative study and provides an example latent space analysis, and finally, Section Discussion and conclusions summarizes our work, discussing potential applications and future research directions.

## Preliminaries

### Drug-target interaction prediction

The advancement of artificial intelligence technologies has significantly impacted various aspects of the medical domain, including processing biological networks [9], diagnosing patients [10], and even accelerating different stages of drug development. In recent years, classical machine learning methods have become increasingly common among DTI prediction approaches as well [3, 11], enabling the estimation of interactions between unknown proteins and molecules. The experimental validation of compound-protein pairs often demonstrates a significant correlation between predicted and measured bioactivities, highlighting the potential of predictive models to fill experimental gaps in existing compound-target interaction maps [12]. This capability proves valuable in early development by identifying candidates that bind to specific proteins or revealing new therapeutic applications for existing drugs through repositioning.

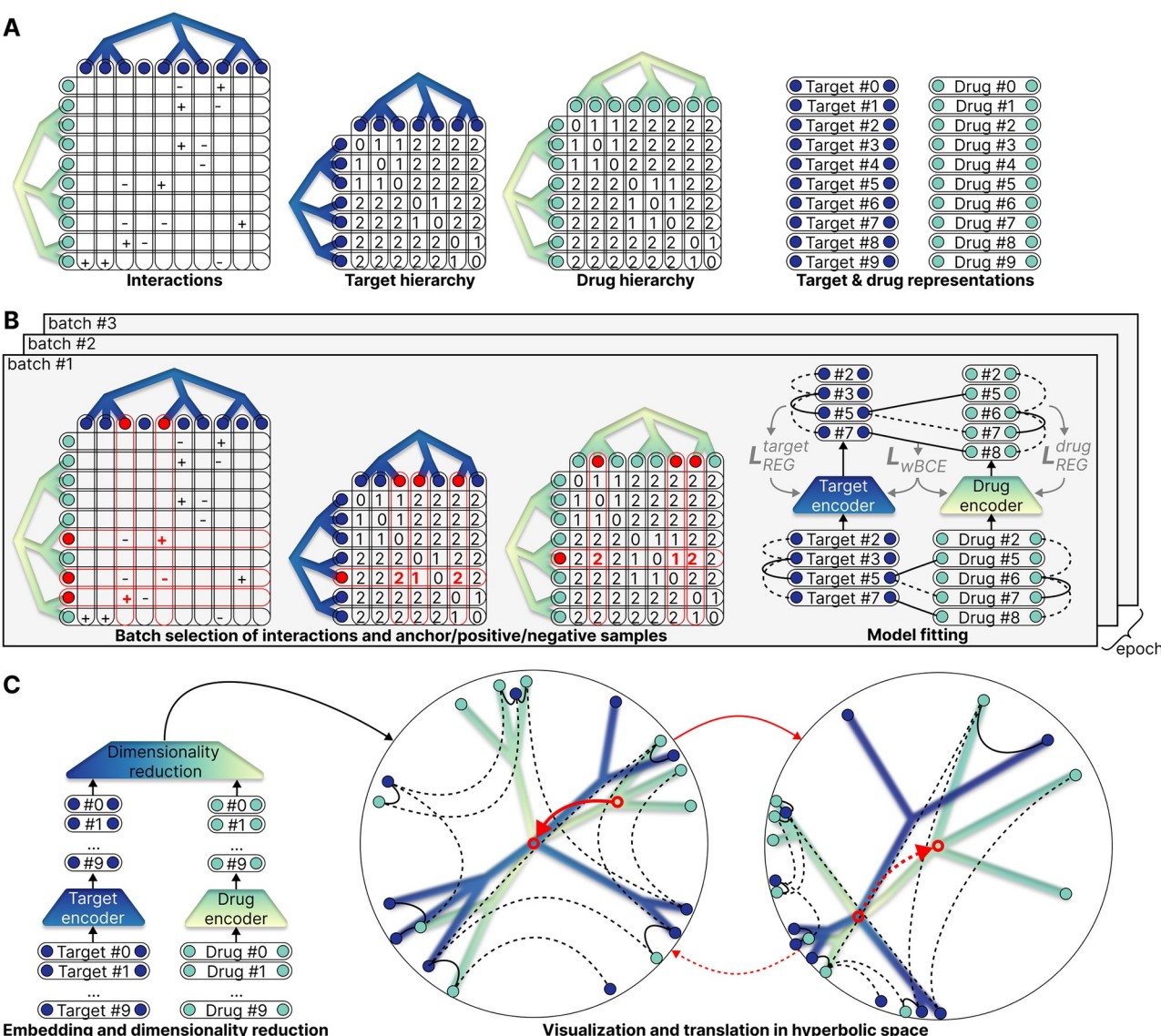

**Fig 1. Overview of the proposed method.** *(A)* As a preprocessing step, the DTI matrix for the positive and negative interactions, the lowest common ancestor distance matrices for the hierarchies, and the input representations are created and stored. *(B)* Then, during the training phase, *b* interactions, *n* drug and target anchors, and for each anchor, one positive and *m* negative samples are selected in each batch (in this example, *b* = 3, *n* = 1, *m* = 2). Utilizing the distances between the model embeddings of the selected samples, the loss is calculated, and based on its gradient, the model weights are updated. *(C)* Finally, the resulting model allows the visualization of the joint-embedding DTI space, which now encodes interactions as well as prior hierarchies. After encoding input representations and subsequently applying dimensionality reduction, the hyperbolic latent space of the model can be visualized in two dimensions. The latent space might be subject to further analysis, such as navigating with the translation operator and identifying clusters of interest. For more details see the following subsections: *(A)* Datasets, *(B)* Model and objective function, *(C)* Latent space analysis.

Most current DTI prediction approaches are based on a *pairwise* neural network model involving dual inputs, such as a molecule and a protein. An illustration of this model type is the DeepDTA architecture [13], which takes a *Simplified Molecular Input Line Entry System* (SMILES) drug representation and an amino sequence on the input. Convolutional encoders process these inputs to generate drug and protein embeddings, which are then concatenated, and a *multi-layer perceptron* (MLP) with a *sigmoid* activation predicts the interaction. An alternative, widely used solution is to utilize pre-trained embeddings on the input, potentially

reducing convergence time while reaching the same predictive performance with less data [14]. Another challenge in DTI prediction is the imbalanced nature of the available datasets. Possible solutions include weighted negative sampling, utilizing unlabeled interaction information in a semi-supervised way [15], or even applying feature selection [16].

Pairwise methods often result in a shared drug-target latent space. In these joint-embedding models, the similarities between the entities reflect interaction information, i.e., interacting pairs are close to each other. This makes pairwise models parallel to some representation-learning approaches. For instance, in a prior study, we showed a way to treat interaction prediction with imbalanced data as a representation learning problem, leveraging the advantages of the state-of-the-art metric learning approaches [17]. Beyond their improved effectiveness, similarity-based models offer a degree of transparency. Visualizing their latent space might provide some explainability by analyzing predicted interactions based on the distances between corresponding entities and identifying clusters with similar drugs or targets nearby.

## Explainable AI

Despite their success, machine learning models are often perceived as black boxes, hindering their interpretability, a critical factor, particularly in medical applications [18]. Therefore, the need for *eXplainable Artificial Intelligence* (XAI) solutions is increasing, aiming to provide transparency and facilitate decision-making. This need has led to the development of various XAI methods, ranging from model-specific to more general, known as model-agnostic approaches [19]. Besides providing interpretable models for machine learning practitioners, there is also a growing demand to increase user trust [20] and even understand how users engage with the explanations [21].

A widespread approach in the medical domain is utilizing interpretable tree-based predictive architectures [18, 22]. Besides, there are existing solutions to enhance the explainability of more complex models, such as deep neural networks, for instance, by generating counterfactual explanations [23, 24] or by attributing the predictions to the input features [25].

XAI approaches are frequently used in drug development to enhance model transparency, identify crucial input features, validate output acceptability, offer insights to decision-makers, and estimate the uncertainty associated with predictions [26]. A prominent example is the utilization of the *SHapley Additive exPlanations* (SHAP) methodology to provide local explanations by identifying compound features responsible for kinase inhibitor activity prediction [27, 28]. Besides, pairwise DTI predictors can also be subjected to XAI solutions, with most recent methods relying on the interpretable-by-design attention mechanism. For instance, some solutions leverage mutual learning attention for drug and target encoders coupled with attention visualization for local interpretation and identification of relevant input features [29]. Others use cross-modal attention with regularization guided by non-covalent interactions [30], and some graph-based models identify protein binding sites for enhanced interpretability [31].

In our current paper, we also use pairwise DTI models. However, instead of local input interpretations provided by the attention mechanism, we aim to make the joint-embedding latent space more explainable by incorporating *a priori* drug and target hierarchies. Besides evaluating the effect of embedding prior biological hierarchies, we extend our investigation to non-Euclidean representations, considering their potential advantages in preserving hierarchy trees.

## Hyperbolic geometry

Mathematical models of non-Euclidean spaces were originally formulated in hyperbolic geometry and later found their way into machine learning. A fundamental representation of

hyperbolic space is the *Poincaré ball* model or its two-dimensional version, the Poincaré disk, often used for visualization purposes. This model was the first to be used by a machine learning approach to learn continuous, hierarchical word representations [4]. The model represents the hyperbolic space as an open d-dimensional unit ball $\mathcal{B}^d = \{x \in \mathbb{R}^d | 1 > \|x\|\}$ with a Riemannian metric. In this model, the *Poincaré distance* $d_\mathcal{P}$ between vectors $\mathbf{x}$ and $\mathbf{y}$ in the unit ball is defined as:

$$d_\mathcal{P}(\mathbf{x}, \mathbf{y}) = \cosh^{-1}\left(1 + 2\frac{\|\mathbf{x} - \mathbf{y}\|^2}{(1 - \|\mathbf{x}\|^2)(1 - \|\mathbf{y}\|^2)}\right) \tag{1}$$

The Poincaré distance within $\mathcal{B}^d$ increases gradually with the Euclidean norms of vectors $\|\mathbf{x}\|$ and $\|\mathbf{y}\|$, making it suitable for embedding trees and data with inherent hierarchies. For example, positioning the root node of a tree at the origin maintains short distances to other nodes, while distances between leaf nodes near the boundary grow rapidly as their embeddings have norms close to one.

Multiple equivalent models of the hyperbolic space exist, such as the *Lorentz model*, frequently used in machine learning applications for its improved numerical stability [32]. The d-dimensional Lorentz (or hyperboloid) model is represented by a hyperboloid manifold $\mathcal{H}^{d,\beta}$ embedded in a d+1-dimensional Euclidean ambient space. $\mathcal{H}^{d,\beta}$ is defined by Eq (2), where $-1/\beta$ is the constant negative curvature of the space and $\langle ., .\rangle_\mathcal{L}$ is the Lorentzian inner product.

$$\mathcal{H}^{d,\beta} = \{\mathbf{x} = (x_0, \ldots, x_d) \in \mathbb{R}^{d+1} | \langle \mathbf{x}, \mathbf{x}\rangle_\mathcal{L} = -\beta, x_0 > 0\}$$

$$\langle \mathbf{x}, \mathbf{y}\rangle_\mathcal{L} = -x_0 y_0 + \sum_{i=1}^{d} x_i y_i <= -\beta \tag{2}$$

The two models are equivalent, with an invertible mapping $h : \mathcal{H}^{d,\beta} \mapsto \mathcal{B}^d$ given by:

$$h(\mathbf{x}) = \frac{(x_1, \ldots, x_d)}{x_0 + 1} \in \mathcal{B}^d \tag{3}$$

Using Eq (3), the Poincaré distance with the Lorentzian inner product can be expressed as $d_\mathcal{P}(h(\mathbf{x}), h(\mathbf{y})) = \cosh^{-1}(-\langle \mathbf{x}, \mathbf{y}\rangle_\mathcal{L})$. We can use other metrics, such as the *squared Lorentzian distance* defined in Eq (4), which satisfies all distance metric axioms except the triangle inequality and is widely employed in machine learning applications [33].

$$d_\mathcal{L}^2(\mathbf{x}, \mathbf{y}) = \langle \mathbf{x} - \mathbf{y}, \mathbf{x} - \mathbf{y}\rangle_\mathcal{L} = -2\beta - 2\langle \mathbf{x}, \mathbf{y}\rangle_\mathcal{L} \tag{4}$$

While the Lorentz model has its stability, other models have other advantages. For example, the Poincaré model is well suited for visualization since it does not require an extra ambient dimension. More than that, it supports useful operators such as translation, i.e., rotating the Poincaré disc to move the origin to a given point while distances between vectors remain intact. As the spatial resolution is amplified near the origin, translation in hyperbolic space can also be considered a "zoom-in" method by moving the area of interest to the center [4]. Other practical models, like the *Klein model*, facilitate other efficient operations in hyperbolic spaces, such as averaging feature vectors. For instance, when working with Poincare embeddings and aiming to calculate their average (referred to as the *Einstein midpoint* in hyperbolic spaces), the process involves converting the embeddings to Klein coordinates, performing the averaging and converting the result back to $\mathcal{B}^d$, as the Einstein midpoint is most efficiently expressed in the Klein model [34].

Researchers have integrated hyperbolic spaces into deep learning methods, leveraging models and operators from differential geometry and developing hyperbolic versions of known neural architectures. Examples include MLP layers with trainable hyperbolic parameters [35], Poincaré *variational autoencoders* (VAE) [36], and graph convolutional neural networks [37]. To optimize the non-euclidean parameters of these models, Riemannian versions of conventional adaptive optimization methods can be employed [38]. In non-Euclidean machine learning applications, operations like the *exponential* and *logarithmic maps* become essential for converting between Euclidean and hyperbolic embeddings. For instance, in the Lorentz model, the exponential map is defined as:

$$exp_\mathbf{x}(\mathbf{v}) = \cosh(\sqrt{\langle \mathbf{v}, \mathbf{v} \rangle_\mathcal{L}})\mathbf{x} + \sinh(\sqrt{\langle \mathbf{v}, \mathbf{v} \rangle_\mathcal{L}})\frac{\mathbf{v}}{\sqrt{\langle \mathbf{v}, \mathbf{v} \rangle_\mathcal{L}}} \tag{5}$$

Here, $exp_\mathbf{x} : \mathcal{T}_\mathbf{x}\mathcal{H}^{d,\beta} \mapsto \mathcal{H}^{d,\beta}$ maps a tangent vector $\mathbf{v} \in \mathcal{T}_\mathbf{x}\mathcal{H}^{d,\beta}$ onto the Lorentz manifold $\mathcal{H}^{d,\beta}$, where $\mathcal{T}_\mathbf{x}\mathcal{H}^{d,\beta} \subseteq \mathbb{R}^d$ denotes the tangent Euclidean subspace at $\mathbf{x} \in \mathcal{H}^{d,\beta}$.

**Embedding biological hierarchies into hyperbolic space.** Various biomedical domains have an underlying hierarchical structure, making hyperbolic embeddings a reasonable choice. For instance, phylogenetic trees are evident examples. Making use of the hierarchical nature of genetic data, Klimovskaia et al. developed a hyperbolic manifold learning method called *Poincaré maps*, leveraging single-cell RNA sequencing measurements to produce two-dimensional representations of cell trajectories on the Poincaré disk [39]. Hyperbolic embeddings can also be used to reconstruct evolutionary relationships, as demonstrated by Macaulay et al., who successfully employed a hyperbolic Markov Chain Monte Carlo method for Bayesian phylogenetic inference [40].

There are other biology-related hierarchies that can enhance machine learning methods, such as the *Anatomical Therapeutic Chemical* (ATC) drug hierarchy. Yu et al. applied a hyperbolic VAE for drug molecules, aligning the latent space according to the ATC hierarchy and using it for drug repositioning tasks [8].

Proteins also exhibit hierarchies, such as the kinase hierarchy [41]. For instance, the similarities based on the hierarchy of protein kinases can be utilized in multi-target DTI prediction methods to find targets with a stronger multi-task effect [42]. The early use cases of manual Poincaré embeddings for cluster identification and hierarchy visualization [43] have evolved with automatic learning methods. For example, the *PoincaréMSA* method automatically produces kinase hierarchy embeddings with the Poincaré maps dimensionality reduction technique using multiple sequence alignment data [44].

**Hyperbolic DTI prediction.** Recent advancements in drug-target interaction prediction include state-of-the-art hyperbolic solutions. Among the first was the work of A. Poleksic, a hyperbolic *matrix factorization* (MF) method [5]. The paper uses a distance-based prediction in a joint-embedding manner. It compares the squared Lorentzian distance with the Euclidean dot-product metric, demonstrating the superior performance of the hyperbolic version. Another recent approach, the *Fully LOrentz Network Embedding* (FLONE) [6], adopts a similar joint-embedding model with Lorentzian squared distance. Instead of a shallow MF, the method incorporates drug-drug and protein-protein similarity-based inputs, employs hyperbolic neural networks for embeddings, and uses disease-aware drug representations to enhance the performance. Hyperbolic methods extend to graph embeddings as well. Lau et al. applied hyperbolic graph neural networks for drug repurposing in Leishmaniasis [7]. Li et al. used hyperbolic knowledge graph embeddings for protein-protein interaction prediction [45]. Zahra et al. leveraged hyperbolic protein-protein interaction network embeddings to identify

efficient drugs inhibiting closely situated proteins and discover potential synergic drugs inhibiting proteins from the same pathway [46].

While existing approaches utilize hyperbolic embeddings for drug or target hierarchy preservation, and others use shared drug-target representations via pairwise neural networks or graph embeddings, none directly incorporate both known drug and target hierarchies into a shared DTI latent space. We only found existing solutions outside the DTI domain that embed hierarchies of two different modalities into the same space, such as using a shallow Poincaré model in a joint-embedding way for words and labels to pair their hierarchies [47]. Our research aims to bridge this research gap by simultaneously embedding drug and target hierarchies in a joint-embedding interaction space, assessing its impact, and demonstrating explainability through use cases.

## Methods

In the subsequent section, we provide an overview of the datasets, models, and evaluation metrics employed in our study.

### Datasets

**Drug-target interactions.**   To assess the effect of hierarchy regularization, we utilized two DTI benchmark datasets containing interaction between compounds and target proteins given by the dissociation constant. One of them has *protein kinases* as targets, called the *Kinase Inhibitor BioActivity* (KIBA) [48] dataset, which is, to the best of our knowledge, one of the largest DTI datasets with rich, labeled hierarchical information. The other, smaller one contains interactions to *nuclear receptors* (NRs), called the *NUclear Receptor Activity* (NURA) [49] dataset.

As for preprocessing, we binarized interactions using a dissociation constant threshold of 3, following the recommendation of Öztürk et al., the authors of the DeepDTA method [13]. We discarded duplicated compounds lacking a SMILES descriptor or containing more than 100 non-H atoms, as well as compounds for which we could not produce an input representation. Table 1 presents the characteristics of the resulting data.

**Input representations.**   In our study, we utilized structure-based input representations generated by pre-trained, unsupervised machine-learning models.

When exploring various molecule representations, including one-hot and categorical inputs without prior information, Morgan fingerprints with various lengths, SMILES descriptor, and the 300-dimensional, pre-trained *Mol2vec* embeddings [50], the latter proved most efficient and achieved superior performance. We produced the embeddings with the pre-trained

**Table 1. Summary of the used datasets.**

|                                                  | KIBA      | NURA      |
|--------------------------------------------------|-----------|-----------|
| Number of compounds                              | 50,418    | 31,006    |
| Compounds with known hierarchy                   | 272       | 42        |
| Number of proteins                               | 467       | 22        |
| Proteins with known hierarchy                    | 409       | 22        |
| Number of interactions                           | 235,625   | 39,162    |
| Number of positive interactions                  | 72,944    | 30,868    |
| Number of negative interactions                  | 162,681   | 8,294     |
| Sparsity (ratio of the unknown interactions)     | 98.9993%  | 94.2589%  |

Mol2vec model using the SMILES descriptor as input, excluding the compounds for which the Mol2vec model could not provide an output.

For protein embeddings, we examined several pre-trained models utilizing amino acid sequences [51]. Among these, *ProtVec* [52], which employs the Word2vec concept and is commonly paired with Mol2vec [14], emerged as a standard choice. While two of the other pre-trained methods, namely *CPCProt* [53] and *ProtTrans* [54], also seemed promising, with the former slightly improving predictive performance and the latter exhibiting better hierarchy preservation, ProtVec offered a balanced trade-off between these considerations. Additionally, we explored alternative approaches, such as using one-hot and categorical inputs, as well as employing bag-of-words representations with k-mers of various lengths in the amino sequences. We also experimented with different dimensionality reduction techniques on the input representations and the combination of different embeddings, utilizing appropriate multimodal encoders. However, none of the methods listed above managed to outperform the pre-trained embeddings. Consequently, we opted for the 300-dimensional ProtVec protein embeddings and used them throughout our study.

**A priori hierarchies.**   Known biological hierarchies for compounds and proteins were obtained and utilized as prior information for model regularization. Notably, for both hierarchies, the entities we want to embed are situated at the leaves, making it unnecessary to learn representations for the internal nodes of the hierarchy trees.

We utilized the ATC hierarchy for drug compounds, precisely the ATC subset provided by Yu et al. [8]. We mapped the hierarchy information to our compounds based on canonical SMILES descriptors, resulting in 272 and 42 compounds with ATC information in the KIBA and NURA datasets. The ATC hierarchy comprises five levels denoted from *ATC1* to *ATC5*, where *ATC1* represents the top level, and *ATC5* includes the leaves (drugs). A distance matrix based on ATC was created to facilitate hierarchy-based representation regularization. As a distance, we chose to use the level of the lowest common ancestor in the ATC tree, ranging from 1 (drugs with the same *ATC4* label) to 5 (drugs with different *ATC1* labels). Note that the leaves are not unique in ATC, i.e., one drug may appear more than once with different hierarchy classes on the higher levels. Duplicated drugs were handled by selecting the corresponding leaf with the lowest distance.

For the protein kinases in the KIBA dataset, we employed the human kinome tree hierarchy defined by Manning et al. [41] and managed to map hierarchy information to 409 targets via UniProt IDs. This four-level hierarchy includes *group*, *family*, *subfamily*, and the genes as leaves. We applied the same lowest common ancestor distance, which ranges from 1 to 4 in this case. Two groups, *other* and *atypical*, were treated differently. If the lowest common ancestor of two proteins is the node belonging to the *other* or *atypical* group, then the distance between them is 4 instead of 3, reflecting the expectation that proteins in these groups are less similar.

We created a similar distance matrix for the nuclear receptors, using the three-level *subfamily-group-member* NR hierarchy known for all 22 targets given by Wang et al. [49].

An abstract overview of the resulting preprocessed data is found in Fig 1A.

## Model and objective function

**Pairwise DTI model.**   We compared pairwise representation-based predictors with squared Lorentz distance and dot product, akin to A. Poleksic [5]. However, instead of a shallow MF with a wrapped normal distribution as prior, we incorporated input representations, similarly to the FLONE method [6]. We chose to work with the commonly used structural

priors, as they enhance the performance and solve the cold-start problem, i.e., the model also works for unseen drugs and targets.

The pre-trained representations, Mol2vec and ProtVec, are inputs to drug and target encoders, i.e., MLPs with hidden ReLU activation and dropout between the layers. Predictions for the binary output interaction are produced through an inner product applied to the resulting drug and target latent representations, followed by a sigmoid activation.

To account for class imbalance, evident in Table 1, we employed the weighted *binary cross-entropy* (BCE) loss as the objective function, denoted as $L_{wBCE}$.

**Hyperbolic version.** To create a hyperbolic version of the predictor, we utilized the Lorentz model and the squared Lorentzian distance as it is widely used in the literature [5, 32, 33]. The output prediction, shaped by an exponential activation, is expressed as $e^{-d_{\mathcal{L}}^2(\mathbf{d},\mathbf{t})}$, where **d** and **t** represent the latent drug and target embeddings, and $d_{\mathcal{L}}^2$ is the squared Lorentzian distance defined by Eq (4). After obtaining latent embeddings with the drug and target encoders, we applied the exponential map from Eq (5) to ensure **d** and **t** are on the Lorentz manifold. Note that we could also use the exponential map on the input Mol2vec and ProtVec representations and use hyperbolic neural networks as encoders, akin to the FLONE method [6]. However, we did not find any increase in performance, and the Riemannian optimization resulted in a much higher computational cost. Consequently, we opted for an Euclidean encoder followed by an exponential map.

A clip regularization was applied to the latent embeddings, ensuring the Euclidean norm of the representations was less than or equal to a pre-defined $\alpha$ threshold. Originally introduced to prevent vanishing gradients during Riemannian optimization [55], we found that even with Euclidean neural networks and the *Adaptive Moment Estimation* (Adam) optimizer, clipping the representations before the exponential map contributed to more stable training.

**Hierarchy regularization.** We incorporated known hierarchies as a priori input to regularize the model embeddings. Following the work of Nickel et al. [32] and Yu et al. [8], we introduced a ranking-based regularization term in the loss function, leveraging pairwise drug-drug and target-target embedding similarities. In batch-wise loss calculation, we first sampled $n$ representations known as anchors (drug or target, depending on the prior hierarchy used). For each anchor $\mathbf{x}_i \in \mathcal{A}$, a positive example $\mathbf{x}_{i+}$ was sampled from the leaves sharing at least one common ancestor with $\mathbf{x}_i$ in the hierarchy tree. Sampling was done uniformly based on the hierarchy-based distance, i.e., the level of the lowest common ancestor between $\mathbf{x}_i$ and $\mathbf{x}_{i+}$. Subsequently, at most $m$ negative samples were randomly selected from the set $\mathcal{K}_i$ consisting of other leaf nodes that are further away in the hierarchy tree from $\mathbf{x}_i$ than $\mathbf{x}_{i+}$. With these samples, the regularization term is expressed as:

$$L_{REG} = \sum_{\mathbf{x}_i \in \mathcal{A}} \log \frac{e^{-d(\mathbf{x}_i, \mathbf{x}_{i+})}}{\sum_{\mathbf{x}_k \in \mathcal{K}_i} e^{-d(\mathbf{x}_i, \mathbf{x}_k)}} \tag{6}$$

We introduced $L_{REG}^{drug}$ and $L_{REG}^{target}$ regularization terms to the final loss function, weighted with $\lambda^{drug}$ and $\lambda^{target}$, the resulting loss function is the following:

$$L = L_{wBCE} + \lambda^{drug} L_{REG}^{drug} + \lambda^{target} L_{REG}^{target} \tag{7}$$

With Eq (7) as the objective function, the model can simultaneously embed drug and target hierarchies while pairing them based on the available interaction information. Fig 1B shows the batch selection and the training process, while Fig 2 depicts an overview of a hyperbolic pairwise model regularized with prior hierarchies.

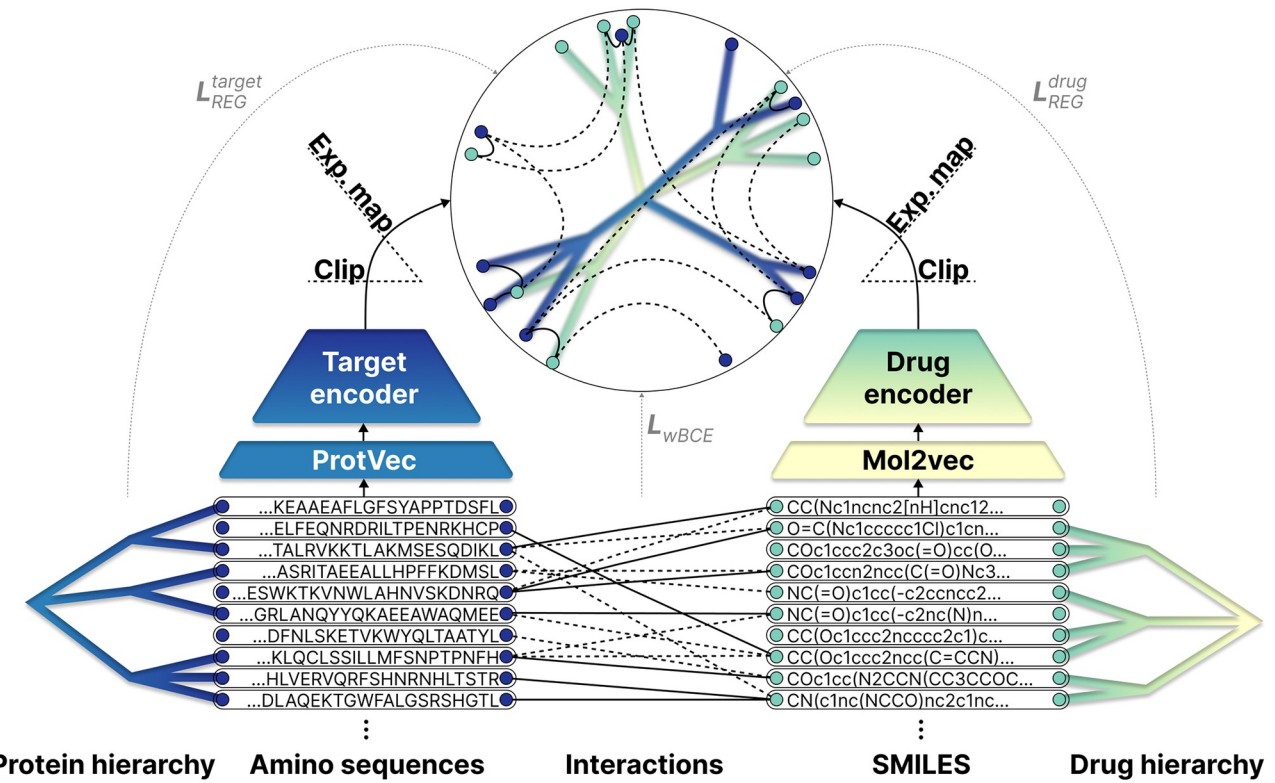

**Fig 2. An illustrative overview of the proposed pairwise architecture using a hyperbolic manifold.** Besides the interactions (solid-positive, dashed-negative), the model takes structural (amino sequences and SMILES descriptors) and hierarchical (ATC drug and kinase/NR protein hierarchies) priors as input. The input sequences are first converted to 300-dimensional vectors using pre-trained Mol2vec and ProtVec embeddings. Subsequently, drug and target encoders, coupled with an embedding clip and exponential map, generate drug and protein latent representations within a shared hyperbolic manifold. The squared Lorentzian distances between these latent embeddings are then utilized for interaction prediction. As a result of simultaneously applying $L_{wBCE}$, $L_{REG}^{drug}$, and $L_{REG}^{target}$ in the objective function, the model encodes interactions as well as prior hierarchies in the joint-embedding space. This means that interacting pairs are close to each other, while negative pairs are distant, and entities closer in the prior hierarchy tree are also closer in the latent space, providing some interpretability. For instance, after applying Poincaré maps dimensionality reduction, we can visually identify clusters in the latent space where subtrees of the prior drug and target hierarchies are closely embedded because of the interacting leaf nodes.

## Evaluation metrics

We evaluated and compared our models according to two aspects: the predictive performance of the binary classification task and how well the resulting latent space preserves prior hierarchies.

**Predictive performance.** For the binary interaction classification task, we employed two metrics summarizing the classifier's performance across varying classification thresholds: the *area under the receiver operating characteristic curve* (ROCAUC) and the *area under the precision-recall curve* (PRAUC), both ranging from 0 to 1.

**Hierarchy preservation.** Following the work of Heller et al. [56] and Yu et al. [8], we utilized the *expected dendrogram purity* (EDP) metric to evaluate hierarchy preservation. EDP assesses how well a learned hierarchy tree $\mathcal{T}$ matches discrete class labels $\mathcal{C}$ on the leaf nodes. Measuring dendrogram purity involves examining pairs of nodes within the same class. For each pair, we identify the smallest subtree containing both nodes and calculate the ratio of leaves in that subtree belonging to the same class as the given pair. Dendrogram purity is the average of these ratios across all possible pairs with the same class labels. The metric ranges from 0 to 1, achieving its maximum when all leaves in each class are part of some pure subtree.

We opted for the more efficient EDP, which samples only $k$ pairs for each class label instead of considering all possible pairs. Based on distances in the latent embedding space, we performed agglomerative hierarchical clustering on the drugs and targets to obtain the learned hierarchy trees $\mathcal{T}_{drug}$ and $\mathcal{T}_{target}$, respectively. To calculate the EDP score for a tree, we averaged the scores for that tree with respect to class labels $\mathcal{C}_i$ on each level $i$ in the given prior hierarchy.

## Results

Computations were executed on a 32GB NVIDIA Tesla V100 GPU, and the models were implemented using the PyTorch framework. Notably, following hyperparameter optimization, we found that the model is robust considering most of its parameters except for the latent dimension, i.e., the size of the drug and target embeddings between which the distances are calculated. Therefore, we applied a previously identified effective configuration consistently throughout our study, focusing our detailed comparison solely on the latent dimension. The other hyperparameters did not significantly affect the results and, consequently, were not subjected to a thorough examination. After Xavier weight initialization, we trained the models for 16 epochs using a batch size of 1024 and Adam optimizer with a learning rate of $5*10^{-5}$. The drug and target encoders had two layers with 1024 neurons each, a 10% dropout, and an output layer with configurable latent dimension. The hyperbolic version employed the Lorentz model with $\beta = 1$, i.e., a constant curvature of -1, and clipped the latent embeddings using a threshold of $\alpha = 1$. We used weighted BCE in the final objective function to offset class imbalances. The ratio of the weights for positive to negative interactions was 2 to 1 for the KIBA and 1 to 2 in the case of the NURA dataset. For the hierarchy regularization terms in the loss function, we used $n = 10$ positive and $m = 256$ negative samples. In the final loss, $\lambda^{drug}$ and $\lambda^{target}$ weights were set to 0.1 when hierarchy regularization was applied and 0.0 otherwise. Lastly, the hyperparameter of the EDP evaluation score, determining the number of samples for each class in each level, was set to $k = 10$.

### Comparative analysis

We conducted a comparative analysis to assess the effect of incorporating prior hierarchies into DTI prediction. Employing five-fold cross-validation with an interaction-based train-test split, we explored the influence of hierarchy regularization in models featuring different manifolds and latent dimensions. It's important to note that, due to the interaction-based train-test split, all drugs and targets are included in the training data. Consequently, unlike the AUC scores, EDP is not evaluated on separate test data (mainly due to the small amount of available data with known hierarchy information). This way, this paper does not focus on the generalization capability of hierarchy preservation, simply on how well hierarchies can be encoded and paired in different joint-embedding models and the consequent impact of this regularization on predictive performance. We distinguished four scenarios concerning the regularization, using only the BCE loss, applying either drug or target hierarchy regularization, and utilizing both drug and target priors. For each scenario, we compared the Euclidean and hyperbolic versions of our pairwise model across different latent sizes: 2, 4, 8, 16, and 32. More precisely, in the Lorentz model, the applied dimensions were one larger since the Lorentz manifold embedded in the ambient Euclidean space is one dimension smaller than the representation itself. We performed cross-validation for all setups and repeated the measurements on the KIBA and NURA datasets. Fig 3 shows the resulting means and standard deviations for all evaluation metrics, as well as the time required to train the model for one epoch on our previously mentioned hardware setup.

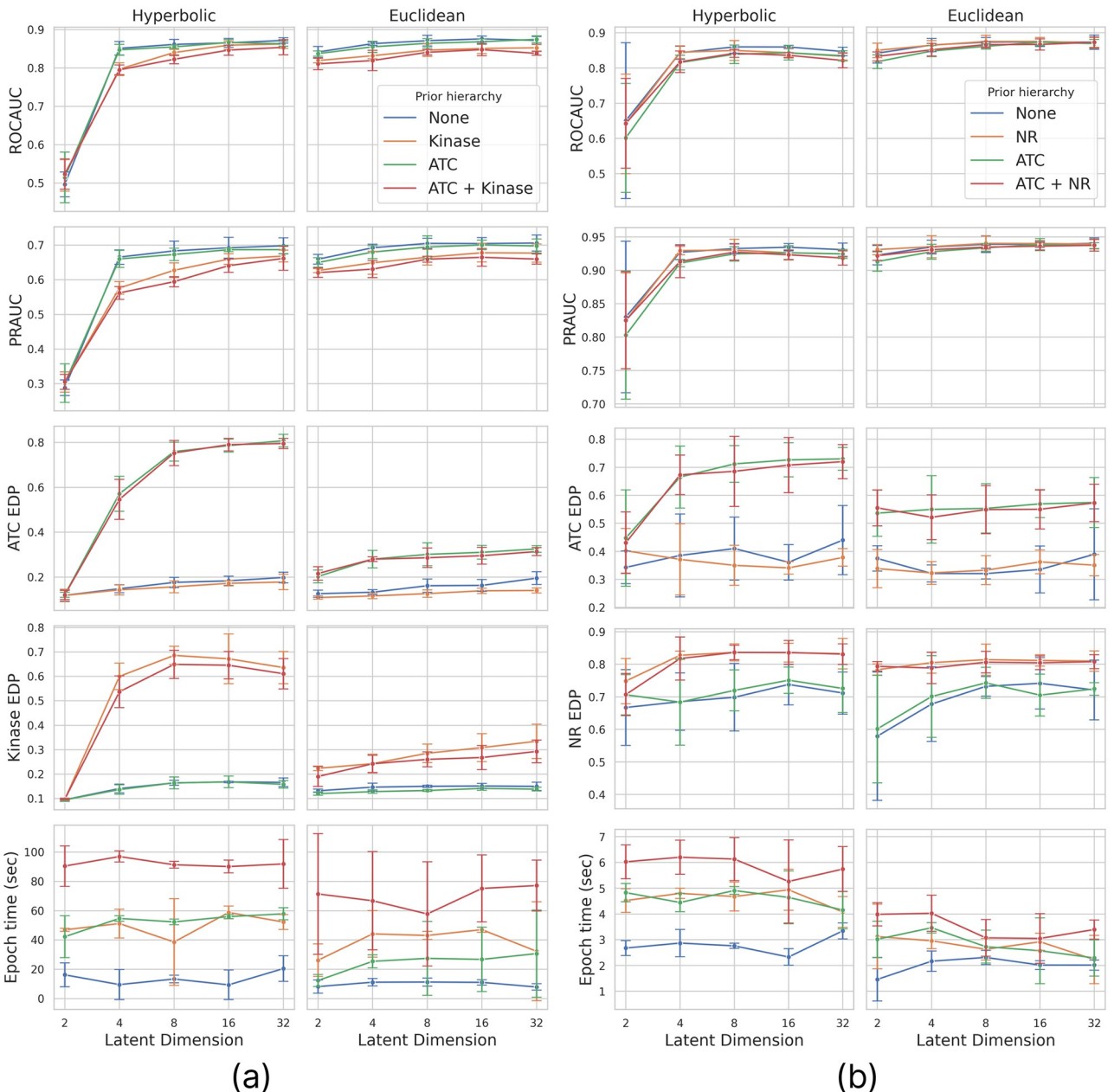

**Fig 3. Cross-validation results on the *(a)* KIBA and *(b)* NURA datasets.** Models with different manifolds are placed in different columns, while the different rows display the various evaluation metrics and the time required to train one epoch. Within each plot, four curves corresponding to four scenarios based on the applied a priori hierarchies depict mean and standard deviation values for different latent dimensions obtained through cross-validation.

In terms of predictive performance, contrary to expectations, Euclidean models slightly outperform hyperbolic ones, especially in lower dimensions. However, the differences are subtle except for the two-dimensional case, where the Lorentz distance exhibits somewhat poorer performance. Our observations also indicate that increasing the latent dimension above 16 does not lead to further enhancement in predictive performance. Additionally, when hierarchy regularization is applied, there is a slight decline in AUC scores, especially for the KIBA

dataset, where a greater number of drugs and targets with known hierarchies results in a stronger regularization. However, this is an acceptable tradeoff if we consider the gained interpretability.

Concerning hierarchy preservation, applying the appropriate regularization significantly increases the EDP score. As expected, this increase is more pronounced in a hyperbolic latent space, where large trees can be embedded with less distortion. Notably, in the case of the NURA dataset, where trees are smaller, the hyperbolic model does not significantly outperform the Euclidean model, particularly for the NR hierarchy, where the 22 leaf nodes can be adequately embedded in Euclidean space as well. Analyzing scenarios without prior hierarchies reveals that a hyperbolic embedding space alone does not guarantee a preserved hierarchy. Both regularization and a hyperbolic space are necessary to achieve a high EDP score. We can also examine how the drug and target hierarchies influence each other. Surprisingly, applying one prior hierarchy does not affect the other, indicating that drug and target hierarchies can be paired and embedded together without compromising each other, but they do not exhibit a synergic effect either, i.e., applying only a drug or target prior does not enhance hierarchy preservation in the other modality.

As for computational cost, not surprisingly, fitting the model on the larger KIBA dataset requires more time. Nevertheless, the same conclusion can be drawn from both datasets. The latent dimension does not significantly affect the training time, i.e., increasing the size of the hidden representations does not have an additional cost. However, utilizing non-Euclidean embeddings has a noticeable overhead, more precisely, a multiplicative cost, with training hyperbolic models for one epoch taking approximately 1.6 times as long as for the Euclidean ones. On the other hand, hierarchy regularization has rather a cumulative cost, with the extra cost of row and column regularizations independently added to the base training time. Notably, after fitting the model, the proposed modifications do not have an extra cost, as the regularization is only applied during training, and the extra exponential map required in the forward pass for the hyperbolic version does not significantly increase inference time.

Based on the results, hierarchy regularization, at the cost of extra training time, contributes to forming a more meaningful embedding space without significantly impacting the predictive performance. Additionally, with more than approximately 30 available leaf nodes in the prior hierarchy tree, using a hyperbolic embedding space is highly recommended, as it enhances the quality of hierarchy preservation at only a small computational cost.

## Latent space analysis

Besides our comparative study, we performed a downstream analysis of the resulting non-euclidean models, showcasing the impact of hierarchy regularization and providing a means to visualize the joint-embedding space. Using a Lorentz manifold and a latent dimension of 8, we trained models applying both drug and target hierarchy regularization with $\lambda^{drug} = \lambda^{target} = 0.5$ weights.

To visualize the hyperbolic latent space, we used the Poincaré disk. With two or three-dimensional models, it is enough to convert the Lorentzian space to Poincare with Eq (3). This is a common approach, also used by Yu et al. [8], to visualize the ATC hierarchy in the VAE latent space. However, we saw that a model with a higher dimension performs better, and we did not want to compromise that just for the sake of visualization. To address this, we applied dimensionality reduction. We tried different hyperbolic methods, including Co-sne [57], but we got the best results with the Poincaré maps [39]. First, we converted the embeddings from the Lorentzian to the Poincaré manifold with Eq (3), then we created a similarity matrix according to the Poincare metric defined in Eq (1). With this pre-calculated similarity matrix

as input, we ran the Poincaré maps for 2000 epochs with a batch size of 64 and a learning rate of 0.05. While keeping the other hyperparameters on their default value, we set the neighbor number to 5 and the output dimension to 2. Utilizing Poincaré maps to embed kinase hierarchies is not a novel concept. In a related method, PoincaréMSA [44], kinase proteins were embedded in an unsupervised way. However, in our approach, instead of multiple sequence alignment data, we utilized ProtVec as input. Another key distinction is that our resulting space is multimodal and jointly optimized to preserve both the ATC and kinase hierarchies as well as DTI information, necessitating the application of hierarchy regularization. An overview of the dimensionality reduction and the visualization process is shown in Fig 1C.

For our analysis, we jointly reduced the drug and target representations with known hierarchy information, i.e., the leaf node in the prior hierarchies. These were visualized in the Poincaré disk and were colored according to class labels on different hierarchy levels. Additionally, we embedded the hierarchy trees bottom-up by iteratively producing the representations of the parent nodes with a hyperbolic average of their children. To calculate the average, we utilized the Klein model. More precisely, we transformed the Poincaré embeddings to the Klein model, calculated the Einstein midpoint, transformed the result back to the Poincaré disk, and connected it to its children. We also visualized the resulting dendrograms after hierarchical clustering to compare the prior hierarchy trees with the ones reconstructed by the model, providing a more interpretable alternative to the dendrogram purity metric. To do so, we used the squared Lorentzian distance matrix between the latent representations and applied hierarchical agglomerative clustering using the farthest point algorithm to obtain a dendrogram. We did the same to the lowest common ancestor-based distance matrix to produce a dendrogram belonging to the prior hierarchy tree. The results are presented in Figs 4 and 5 for the KIBA and NURA datasets.

Analyzing the latent representations reveals their alignment with the hierarchies, an observation supported by EDP scores as well. The KIBA model achieves EDP scores of 0.7669 and 0.8301 for the ATC and kinase hierarchies, respectively, indicating slightly better preservation of the kinase hierarchy. We can confirm this by visually inspecting the two-dimensional tree embeddings in Fig 4. The root is indeed close to the origin, while the leaves are situated near the boundary of the Poincaré ball, with edges of the hierarchy tree rarely crossing each other. Notably, most of the crossings are associated with the two group-level modes being close to the root, the *other* and *atypical* groups, for which we did not regularize the embeddings to be close to each other (unless they were part of the same family or subfamily). For all the other groups, the corresponding embeddings are close to each other, meaning that their Einstein midpoint is close to the boundary, i.e., the group-level embeddings are close to the boundary, resulting in fewer crossings between the edges. The EDP scores with the NURA model are 0.6796 for the drugs and 1.0 for the NRs. An EDP score of 1.0 means that all targets belonging to the same subfamily or group are contained in a pure subtree. In Fig 5, we can see that the NR tree is embedded without crossing edges even after the dimensionality reduction. However, this is an easy task as the hierarchy has only 22 leaf nodes.

As an attempt to elucidate the latent space, we investigated how the two hierarchies are positioned relative to each other, i.e., identified formations consisting of hierarchy clusters of different modalities. Fig 4 shows such an example with the cluster of *tyrosine kinases* (TK), *antineoplastic and immunomodulating agents*, and *musculoskeletal system drugs* positioned close to each other in a specific region of the Poincaré disk. The relationship between the target and the two drug groups is indeed known in the literature. TK enzymes are overactive or found at high levels in certain cancer cells. Therefore, inhibiting them can prevent these cells from growing [58]. It has also been published that the discontinuation of some TK inhibitor-based therapies can lead to musculoskeletal pain as a withdrawal problem [59]. Furthermore,

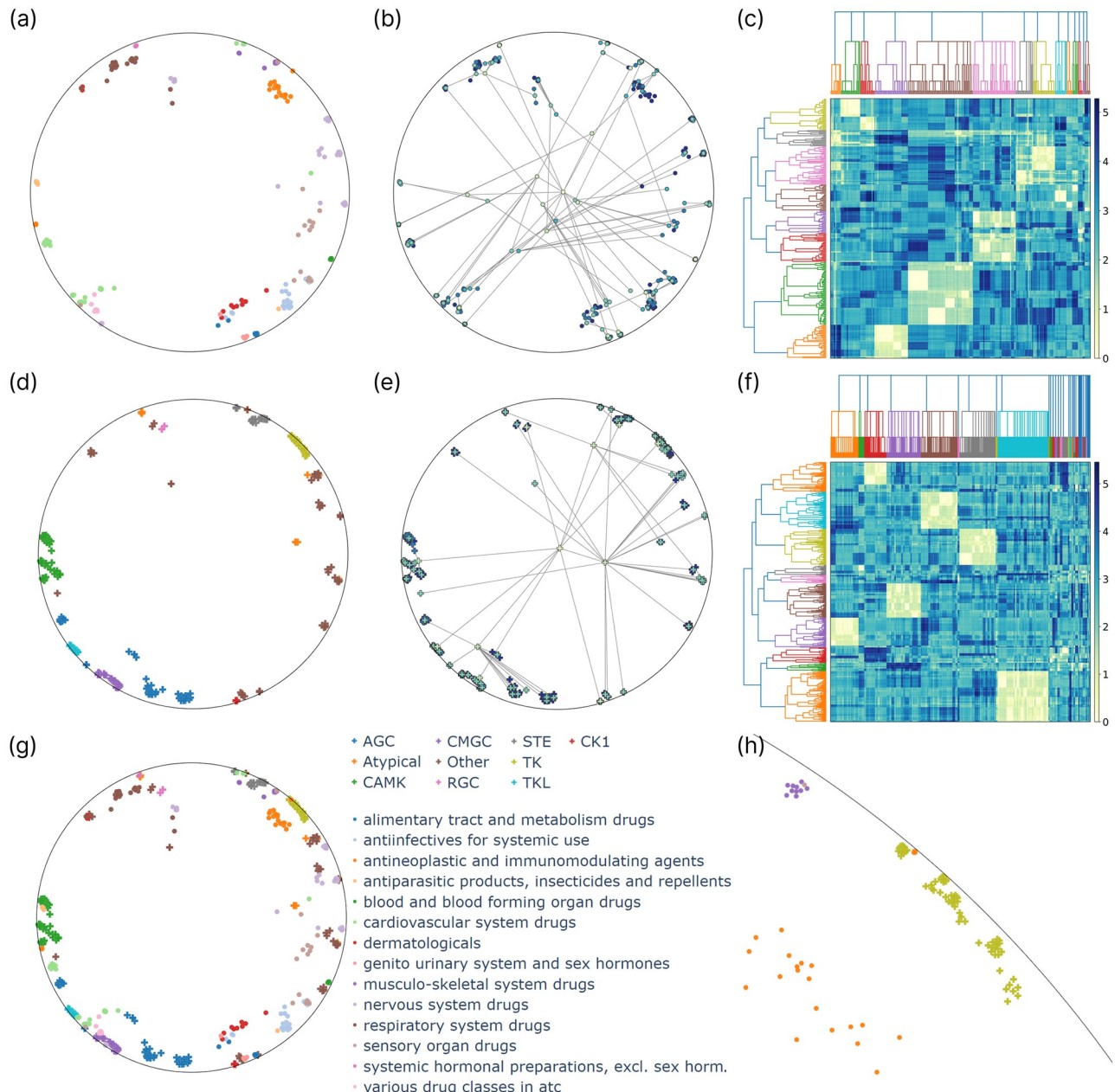

**Fig 4. Analysis of the KIBA latent space, with embeddings reduced via Poincaré maps and shown in the Poincaré disk.** First, drugs *(a-c)* and targets *(d-f)* are analyzed separately, with drugs colored by ATC1, the top level of the ATC hierarchy *(a)* and proteins colored by kinase groups *(d)*. The embedded ATC *(b)* and kinase hierarchy trees *(e)* are presented with nodes colored by hierarchy level. As a visual alternative to dendrogram purity, heatmaps show the drug *(c)* and target similarity matrices *(f)*, which are ordered along the prior dendrogram (top) and the result of hierarchical clustering (left). Drug and target embeddings are also shown jointly in the shared latent interaction space *(g)*, with a highlighted section of the Poincaré disk *(h)* illustrating an example arrangement of clusters belonging to different modalities.

there is evidence that proteins in the ROR family of the TK group play a crucial role in morphogenesis and formation of the musculoskeletal system during embryonic development as well as the regeneration and maintenance of the musculoskeletal system in adults [60]. In other words, we can explain why these three clusters are close in the latent space. Additionally, in the case of unknown drugs and targets that come close to them, we can assume that they are

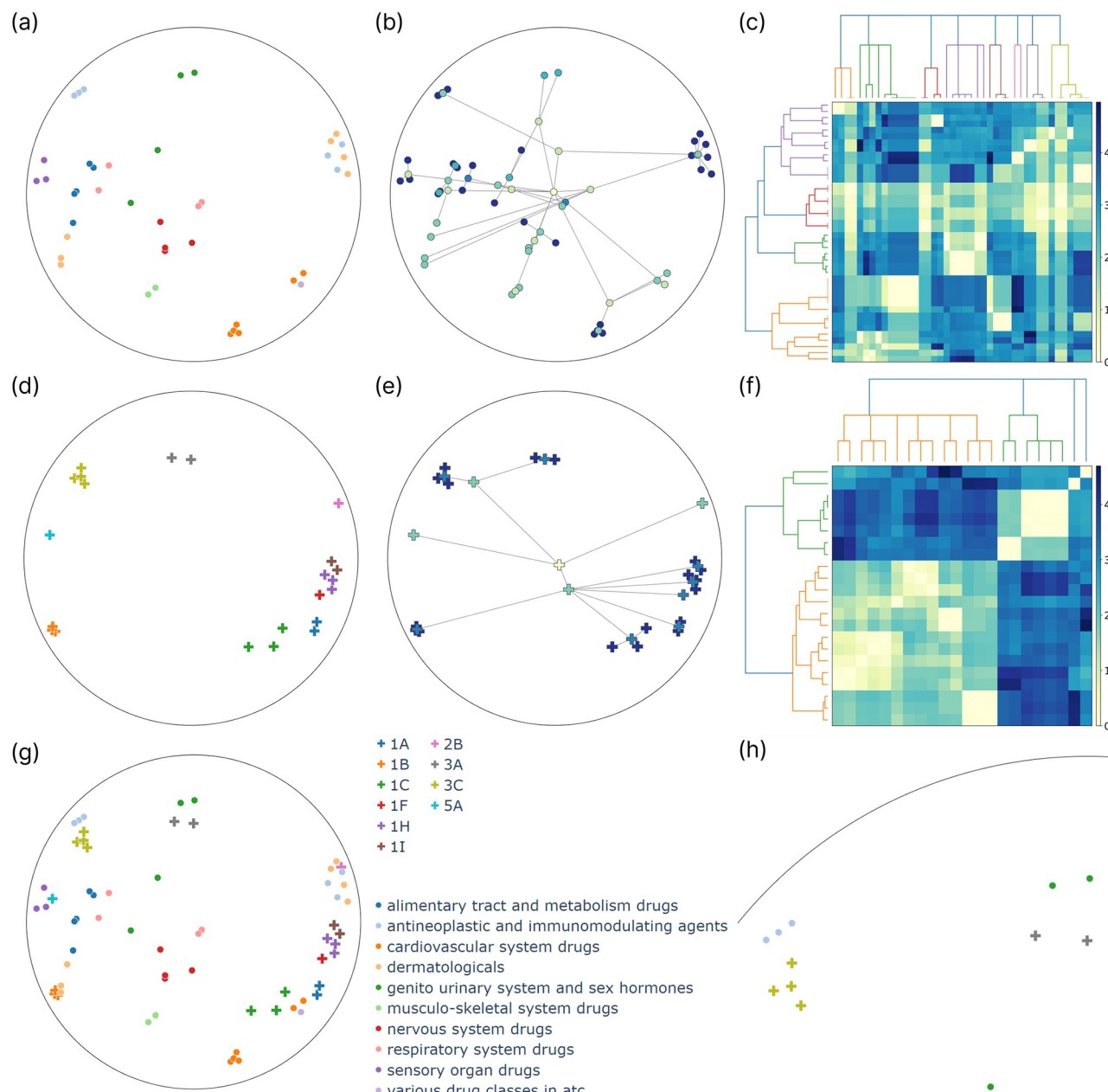

**Fig 5. Analysis of the NURA latent space, with embeddings reduced via Poincaré maps and shown in the Poincaré disk.** First, drugs *(a-c)* and targets *(d-f)* are analyzed separately, with drugs colored by ATC1 *(a)* and proteins by the top two levels of the NR hierarchy, the subfamily and group *(d)*. The embedded ATC *(b)* and NR hierarchy trees *(e)* are presented with nodes colored by hierarchy level. As a visual alternative to dendrogram purity, heatmaps show the drug *(c)* and target similarity matrices *(f)*, which are ordered along the prior dendrogram (top) and the result of hierarchical clustering (left). Drug and target embeddings are also shown jointly in the shared interaction space *(g)*, with a highlighted section of the Poincaré disk *(h)* illustrating an example arrangement of clusters belonging to different modalities.

somehow related to the growth of cancer cells and the maintenance of the musculoskeletal system. Fig 5 shows a similar analysis with the NURA model, revealing a region of the Poincaré disk where targets belonging to the NR subfamily 3 are embedded. It can be further divided into two parts according to the next hierarchy level, the groups, one with the *NR3A* and another with the *NR3C* group. There are two clusters of drugs in this region, the *antineoplastic*

*and immunomodulating agents* and the *genito urinary system and sex hormones*, the former being closer to the *NR3C* and the latter to the *NR3A* group. This formation is also consistent with previous literature. For instance, the second member of the *NR3C* group plays a tumor-suppressive role in colon cancer [61], while the first member, *NR3C1*, is involved in multiple sclerosis, which is an immune-related disease [62]. *NR3A* and *NR3C* are both steroid receptor groups. Among them, the members of *NR3A* are known estrogen receptors present, for instance, in Leydig cells [63]. This analysis uncovers latent space regions associated with cancer, immune response, and hormonal regulation.

The previous examples demonstrated the visualization of hierarchy trees and the intertwining of hierarchies of the two different modalities. Nevertheless, we have not yet fully exploited the possibilities of the Poincaré disk visualization. Fig 6 presents another use-case with the KIBA model utilizing the translation operator to navigate the hyperbolic space. The used model representations and the color of the nodes are the same as in Fig 4. Still, notably, due to the nondeterministic nature of Poincaré maps dimensionality reduction, the 2-dimensional visualizations are not precisely the same. We can see that by rotating the disk, substructures with their neighborhood can be examined more thoroughly on different hierarchy levels. This

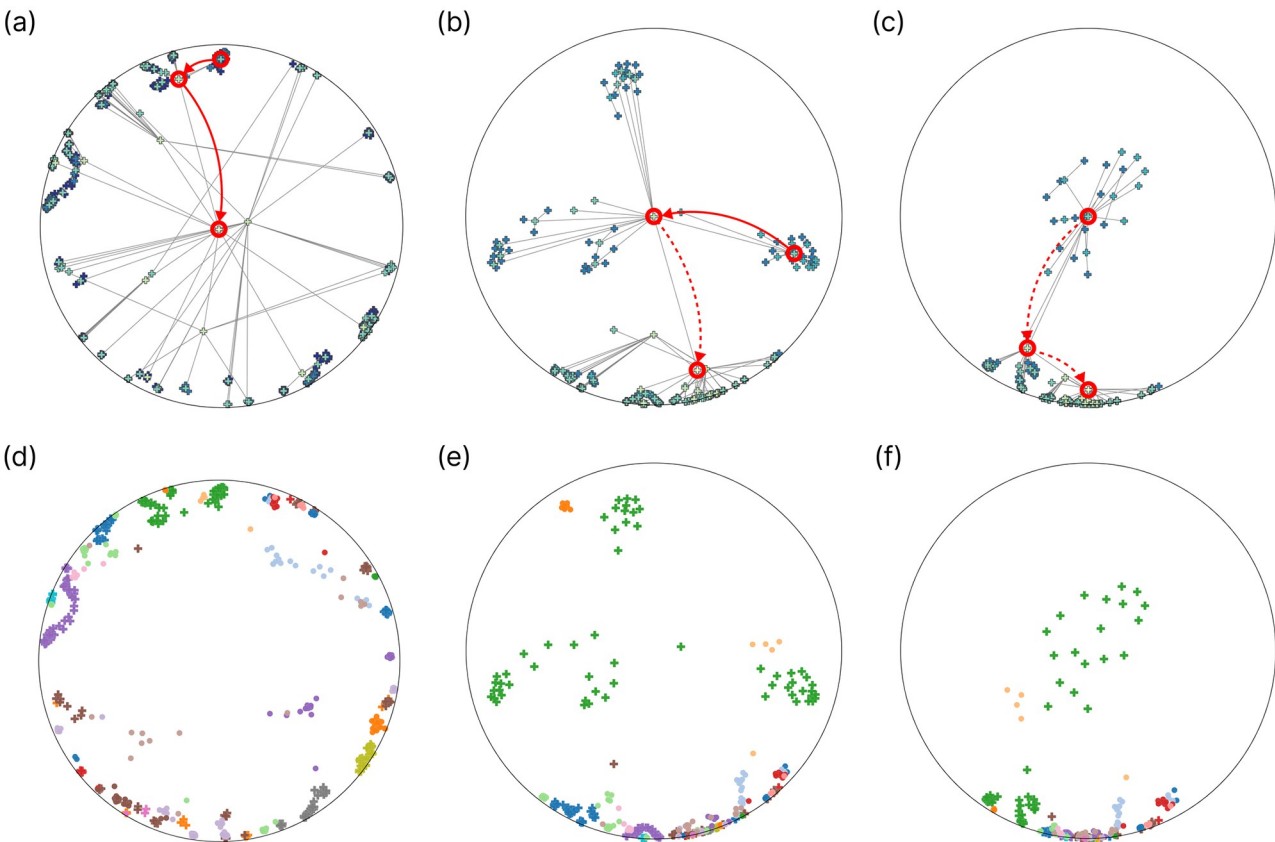

**Fig 6. Navigating the KIBA latent space by the translation operator on the Poincaré disk.** In the top row *(a-c)*, the kinase hierarchy tree is depicted, with nodes colored by hierarchy level, while below *(d-f)*, the jointly embedded drugs and targets are shown, colored by ATC1 and kinase groups, respectively. From left to right, the figure illustrates the translation process through different nodes in the hierarchy tree. Solid and dashed red arrows indicate translations yet to occur and those already completed. In the first column *(a, d)*, the original result of the Poincaré maps dimensionality reduction is presented. In the middle *(b, e)*, the internal node of the hierarchy tree corresponding to the *CAMK* group, i.e., the average of the representations belonging to that group, is translated to the origin. Finally, the last column *(c, f)* displays the neighborhood of the *CAMKL* family after translating the corresponding node of the tree to the center.

is possible because translating the internal nodes of the hierarchy tree to the origin where distances are perceived to be larger allows the observation of fine details.

## Discussion and conclusions

In the paper, we extensively compared Euclidean and non-Euclidean models, incorporating various prior hierarchies and latent dimensions. Contrary to our expectations, using a pairwise model with pre-trained input representations, Euclidean versions perform similarly or even slightly better according to the binary classification task and are computationally more efficient. Our experiments revealed that hierarchy regularization does not enhance predictive performance but significantly improves dendrogram purity without compromising the predictions. Additionally, applying regularization in combination with a hyperbolic space leads to an ample increase in the quality of the embedded hierarchy trees. Considering these insights, if the primary objective is to optimize the predictive performance of pairwise DTI models, opting for a more efficient Euclidean model with input representations might be the preferred choice. However, if creating an interpretable latent space is also a priority, hyperbolic embeddings are reasonable alternatives. It is important to note that while hyperbolic embeddings enhance interpretability, they do not eliminate the necessity for hierarchy regularization. Or at least in the case of the tried input representations, without regularization, the resulting EDP scores are similar for the hyperbolic and Euclidean models. Although the method is easy to implement, robust to the hyperparameters, and does not introduce significant constraints regarding the model architecture, it increases complexity and training time. As a rule of thumb, over approximately 30 leaf nodes to embed, we recommend using hyperbolic space with the regularization as it significantly increases the hierarchy preservation without sacrificing predictive performance.

We also performed a visual analysis to show how hyperbolic models can provide insights into the latent structures when paired with hierarchy regularization and subsequent Poincaré maps dimensionality reduction. As an example, we investigated regions of the space where hierarchy subtrees with different modalities are closely embedded. Nevertheless, Poincaré maps visualization of the regularized latent space offers a wide variety of experiments beyond this particular analysis. One option is to reduce embeddings for all drugs and targets, not just those with a known prior hierarchy, and predict hierarchy labels on a chosen level using latent space similarities. Another possibility is to embed new drugs or targets with given Mol2vec and ProtVec representations, visualize their embeddings in the Poincaré disk, and explore potential interactions and hierarchy labelings based on observed similarities. For instance, akin to the NR-2L method [64], we can use the NURA model to identify a nuclear receptor subfamily or group to which a query protein is most similar. Besides the new opportunities, visualizing the latent space raises new challenges when applied in practical drug discovery pipelines. A key concern requiring further investigation is the tradeoff between losing interpretability with dimensionality reduction and losing expressive power with a small latent dimension.

The regularization can be extended to handle hierarchies where the internal nodes are also represented. For example, when dealing with drug repositioning datasets where diseases serve as targets, the *Medical Subject Headings* (MeSH) [65] can be employed instead of protein hierarchies. Unlike protein hierarchies, MeSH incorporates targets not only at the leaves but also within internal nodes, i.e., it is possible that one target is a descendant of another. The only modification needed in our method is to replace the lowest common ancestor distance with the number of hops in the shortest path between two diseases. With these adjustments, we ran the model on a subset of the ChEMBL [66] dataset introduced in our previous

study [17]. Utilizing the MeSH and ATC as prior hierarchies, we obtained similar results, achieving a high EDP score without significantly impacting predictive performance. Our experiences indicate the robustness of the regularization method across various hierarchy types. Including the NR with a regular tree structure, the kinase hierarchy with the other and atypical groups treated differently, the ATC with leaves belonging simultaneously to multiple labels in different hierarchy levels, and the MeSH where internal of the tree are also embedded. This opens the possibility to experiment with different types of hierarchies, potentially identifying new modalities with corresponding hierarchies that can even increase the predictive performance.

Although the regularization approach is not model-agnostic, it is not highly restrictive either, being compatible with pairwise DTI methods. Further transparency can be achieved by applying other existing XAI solutions, such as utilizing mutual learning attention layers in drug and target encoders and visualizing input feature importance based on attention weights [29]. Alternatively, the SHAP [27, 28] or other gradient or perturbation-based methods can also be employed to determine feature importance [25], i.e., parts of the drug and target structure responsible for the given prediction. The regularization is also compatible with the FLONE method [6], making it possible to handle disease-specific drug representations. In addition, akin to the work of Zahra et al. [46], our method can be used for informed drug repositioning guided by the embedded hierarchies.

We can use low-complexity model alternatives as well, for instance, in replicating the work of A. Poleksic [5], employing a shallow matrix factorization without structural prior inputs, we confirmed that MF with only one-hot or categorical input representations benefits from the hyperbolic distance, achieving superior predictive performance compared to the Euclidean version, yet it also required supervised regularization to embed hierarchies. However, utilizing the pre-trained inputs, our architecture performs better than both versions of the shallow MF. As we showed, with this extra model complexity, the model no longer benefits from a hyperbolic embedding space. Nonetheless, we believe that hyperbolic embeddings can extract additional information when employed with a suitable input modality instead of pre-trained representations produced with a Euclidean approach.

Future research efforts should identify input representations intrinsically suited for hyperbolic space, such as transcriptomic data or structure-based pre-trained embeddings produced with an appropriate hyperbolic method. Given our observation that different embeddings yield significantly different performances, utilizing suitable input representations might enhance the predictive performance of hyperbolic DTI models and potentially eliminate the need for hierarchy regularization. Another challenge to be addressed is the limited availability of public data containing abundant, labeled hierarchies, leading to the need for diverse DTI benchmarks and datasets focusing on biological hierarchies.

In conclusion, our joint-embedding hierarchy regularization, compatible with joint-embedding methods, results in a more interpretable and meaningful latent space with minimal impact on the performance, hopefully facilitating a more informed and insightful application of pairwise DTI prediction models in drug discovery.

## Acknowledgments

The authors would like to thank András Millinghoffer for carefully preprocessing and providing the KIBA and NURA interaction datasets.

## Author Contributions

**Conceptualization:** Domonkos Pogány, Péter Antal.

**Funding acquisition:** Péter Antal.

**Investigation:** Domonkos Pogány.

**Methodology:** Domonkos Pogány.

**Software:** Domonkos Pogány.

**Supervision:** Péter Antal.

**Validation:** Domonkos Pogány, Péter Antal.

**Visualization:** Domonkos Pogány.

**Writing – original draft:** Domonkos Pogány.

**Writing – review & editing:** Domonkos Pogány, Péter Antal.

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
