## [Decision Letter · Decision Letter 0]

2 Jan 2024

PONE-D-23-41591Towards explainable interaction prediction: Embedding biological hierarchies into hyperbolic interaction spacePLOS ONE

Dear Dr. Pogány,

Thank you for submitting your manuscript to PLOS ONE. After careful consideration, we feel that it has merit but does not fully meet PLOS ONE’s publication criteria as it currently stands. Therefore, we invite you to submit a revised version of the manuscript that addresses the points raised during the review process.

We look forward to receiving your revised manuscript.

Kind regards,

Xiao Luo

Academic Editor

PLOS ONE

Journal Requirements:

Reviewers' comments:

Reviewer's Responses to Questions

**Comments to the Author**

1. Is the manuscript technically sound, and do the data support the conclusions?

Reviewer #1: Yes

Reviewer #2: Yes

2. Has the statistical analysis been performed appropriately and rigorously? 

Reviewer #1: Yes

Reviewer #2: Yes

3. Have the authors made all data underlying the findings in their manuscript fully available?

Reviewer #1: Yes

Reviewer #2: Yes

4. Is the manuscript presented in an intelligible fashion and written in standard English?

Reviewer #1: Yes

Reviewer #2: Yes

5. Review Comments to the Author

Reviewer #1: The work aims to enhance the interpretability of machine learning models in drug discovery by embedding these hierarchies into a joint-embedding latent space and comparing it with traditional Euclidean models. The manuscript utilizes two benchmark datasets, KIBA and NURA, for evaluating the models. Predictive performance is measured using the area under the receiver operating characteristic curve (ROCAUC) and the area under the precision-recall curve (PRAUC). Hierarchy preservation is assessed using the expected dendrogram purity (EDP) metric.

Strengths

1. Innovative Approach: The manuscript successfully integrates biological hierarchies into the interaction prediction process, addressing a critical gap in explainable AI within drug discovery.

2. Comparative Analysis: The comparison between hyperbolic and Euclidean models offers valuable insights into the effectiveness of different embedding techniques in handling hierarchical data.

3. Practical Application: The research directly contributes to enhancing the interpretability of drug-target interaction models, a crucial aspect in the medical field.

4. Comprehensive Evaluation: Employing both predictive performance metrics and hierarchy preservation assessment provides a thorough understanding of the model's effectiveness.

Questions

1. How does the model perform when subjected to more diverse and complex datasets beyond KIBA and NURA?

2. Are there other embedding techniques that were considered but not included in this study, and why?

3. How does the model's performance vary with different types of biological hierarchies?

4. What are the potential challenges in implementing these techniques in practical drug discovery pipelines?

5. The article mainly discusses biological network and medical domains. To reflect the frontier and integrity of related work, it is recommended to introduce related methods. [1] KGNN: Harnessing Kernel-based Networks for Semi-supervised Graph Classification. WSDm 2022 [2] Building Conversational Diagnosis Systems for Fine-grained Diseases using Few Annotated Data. ICONIP 2022

Reviewer #2: In this paper, the author propose an explanation hyperbolic-embedding method to explain the drug-target interaction. They try to use the hyperbolic representation to capture the hierarchical information contained in the interactions. Also, they provide a detailed introduction of the hyperbolic method. The innovation is good and the experiment is abundant.

Strength:

1 An exploration to the hyperbolic embedding based explanation of drug-target interaction.

2 The method is plausible and practical. Besides, the intuition to capture the tree structure information is innovative.

3 The experiment is abundant.

Weakness:

1 The author claim that they want to capture the hierarchical information by using hyperbolic embedding. However, they didn’t provide explanation cases in that way. I suggest the author to show a case study to enhance its explainability..

2 The introduction of the algorithm is not that clear. I suggest the author to add a pipeline plot to show the procedure of their algorithm.

3 Although they show the performance in the experiment part, they fail to show the sensitivity of the proposed algorithm. I suggest the author to add an ablation to show its robustness.

4 I suggest the author to analyze the efficiency of the proposed algorithm.

5 Some writting could be polished. I suggest the author to further polish the paper and keep it concise.

6 I suggest the author to add a section to introduce the current progress of XAI. I list some relevant papers as follow:

1 Guerdan, Luke, Alex Raymond, and Hatice Gunes. "Toward affective XAI: facial affect analysis for understanding explainable human-ai interactions." In Proceedings of the IEEE/CVF International Conference on Computer Vision, pp. 3796-3805. 2021.

2 Weitz, Katharina, Dominik Schiller, Ruben Schlagowski, Tobias Huber, and Elisabeth André. "" Do you trust me?" Increasing user-trust by integrating virtual agents in explainable AI interaction design." In Proceedings of the 19th ACM International Conference on Intelligent Virtual Agents, pp. 7-9. 2019.

3 Chen, Ziheng, Fabrizio Silvestri, Jia Wang, He Zhu, Hongshik Ahn, and Gabriele Tolomei. "Relax: Reinforcement learning agent explainer for arbitrary predictive models." In Proceedings of the 31st ACM International Conference on Information & Knowledge Management, pp. 252-261. 2022.

4 Goyal, Yash, Ziyan Wu, Jan Ernst, Dhruv Batra, Devi Parikh, and Stefan Lee. "Counterfactual visual explanations." In International Conference on Machine Learning, pp. 2376-2384. PMLR, 2019.

5 Biran, Or, and Courtenay Cotton. "Explanation and justification in machine learning: A survey." In IJCAI-17 workshop on explainable AI (XAI), vol. 8, no. 1, pp. 8-13. 2017.

6 Yarkoni, Tal, and Jacob Westfall. "Choosing prediction over explanation in psychology: Lessons from machine learning." Perspectives on Psychological Science 12, no. 6 (2017): 1100-1122.

7 Parimbelli, Enea, Tommaso Mario Buonocore, Giovanna Nicora, Wojtek Michalowski, Szymon Wilk, and Riccardo Bellazzi. "Why did AI get this one wrong?—Tree-based explanations of machine learning model predictions." Artificial Intelligence in Medicine 135 (2023): 102471.

8 Lundberg, Scott M., Gabriel Erion, Hugh Chen, Alex DeGrave, Jordan M. Prutkin, Bala Nair, Ronit Katz, Jonathan Himmelfarb, Nisha Bansal, and Su-In Lee. "From local explanations to global understanding with explainable AI for trees." Nature machine intelligence 2, no. 1 (2020): 56-67.

6. PLOS authors have the option to publish the peer review history of their article (what does this mean?). If published, this will include your full peer review and any attached files.

Reviewer #1: No

Reviewer #2: No

---

## [Author Response · Author response to Decision Letter 0]

14 Feb 2024

The authors would like to thank the editor and reviewers for their time and effort dedicated to providing constructive comments that have helped improve the quality of this manuscript. The manuscript has been thoroughly revised according to the reviewers’ insightful comments and questions.

In the marked-up copy of our manuscript (including the figures), we highlighted all the changes made to the original version.

We addressed all questions and comments in a separate response letter.

---

## [Decision Letter · Decision Letter 1]

6 Mar 2024

Towards explainable interaction prediction: Embedding biological hierarchies into hyperbolic interaction space

PONE-D-23-41591R1

Dear Dr. Domonkos Pogány,

We’re pleased to inform you that your manuscript has been judged scientifically suitable for publication and will be formally accepted for publication once it meets all outstanding technical requirements.

Kind regards,

Xiao Luo

Academic Editor

PLOS ONE

Additional Editor Comments (optional):

Reviewers' comments:

Reviewer's Responses to Questions

**Comments to the Author**

1. If the authors have adequately addressed your comments raised in a previous round of review and you feel that this manuscript is now acceptable for publication, you may indicate that here to bypass the “Comments to the Author” section, enter your conflict of interest statement in the “Confidential to Editor” section, and submit your "Accept" recommendation.

Reviewer #1: All comments have been addressed

Reviewer #2: All comments have been addressed

2. Is the manuscript technically sound, and do the data support the conclusions?

Reviewer #1: Yes

Reviewer #2: Yes

3. Has the statistical analysis been performed appropriately and rigorously? 

Reviewer #1: Yes

Reviewer #2: Yes

4. Have the authors made all data underlying the findings in their manuscript fully available?

Reviewer #1: Yes

Reviewer #2: Yes

5. Is the manuscript presented in an intelligible fashion and written in standard English?

Reviewer #1: Yes

Reviewer #2: Yes

6. Review Comments to the Author

Reviewer #1: The author carefully addressed each of my questions, and I am inclined to accept this manuscript to plos one.

Reviewer #2: The author finished polishing the paper and resolved all my concerns. It is ready to get published and I recommend to accept it.

7. PLOS authors have the option to publish the peer review history of their article (what does this mean?). If published, this will include your full peer review and any attached files.

Reviewer #1: No

Reviewer #2: No

---

## [Editor Report · Acceptance letter]

11 Mar 2024

PONE-D-23-41591R1 

PLOS ONE

Dear Dr. Pogány, 

I'm pleased to inform you that your manuscript has been deemed suitable for publication in PLOS ONE. Congratulations! Your manuscript is now being handed over to our production team.

Kind regards, 

on behalf of

Dr. Xiao Luo 

Academic Editor

PLOS ONE